# Depression and Obesity: Analysis of Common Biomarkers

**DOI:** 10.3390/diseases8020023

**Published:** 2020-06-14

**Authors:** Walter Milano, Paola Ambrosio, Francesca Carizzone, Valeria De Biasio, Walter Di Munzio, Maria Gabriella Foia, Anna Capasso

**Affiliations:** 1U.O.S.D. Eating Disorder Unit, Mental Health Department ASL Napoli 2 Nord, 80027 Napoli, Italy; wamilano@tin.it (W.M.); p-ambrosio@libero.it (P.A.); francesca.carizzone@aslnapoli2nord.it (F.C.); valeria.debiasio@aslnapoli2nord.it (V.D.B.); 2Mental Health Department ASL Napoli 2 Nord, 80027 Napoli, Italy; walter.dimunzio@aslnapoli2nord.it (W.D.M.); mariagabriella.foia@aslnapoli2nord.it (M.G.F.); 3Department of Pharmacy, University of Salerno, Fisciano, 84084 Salerno, Italy

**Keywords:** biomarkers, depression, obesity

## Abstract

Depression and obesity are very common pathologies. Both cause significant problems of both morbidity and mortality and have decisive impacts not only on the health and well-being of patients, but also on socioeconomic and health expenditure aspects. Many epidemiological studies, clinical studies and meta-analyses support the association between mood disorders and obesity in relationships to different conditions such as the severity of depression, the severity of obesity, gender, socioeconomic status, genetic susceptibility, environmental influences and adverse experiences of childhood. Currently, both depression and obesity are considered pathologies with a high-inflammatory impact; it is believed that several overlapping factors, such as the activation of the cortico-adrenal axis, the exaggerated and prolonged response of the innate immune system and proinflammatory cytokines to stress factors and pathogens—as well as alterations of the intestinal microbiota which promote intestinal permeability—can favor the expression of an increasingly proinflammatory phenotype that can be considered a key and common phenomenon between these two widespread pathologies. The purpose of this literature review is to evaluate the common and interacting mechanisms between depression and obesity.

## 1. Introduction

Obesity and depressive disorder are two of the most common diseases worldwide and represent considerable problems not only in terms of their strong impact on the health and well-being of individuals, but also for socioeconomic aspects. Furthermore, the prevalence of these two pathologies is growing all over the world. Epidemiological evidence has identified solid associations between depression and obesity [1]. Although distinguishable in terms of etiopathogenetic processes, growing evidence suggests complex two-way relationships between adiposity and depression, which may explain their similar and parallel growth. Depression is associated with an increased risk of weight gain and obesity, which in turn, are associated with a greater vulnerability for depressive disorders [2,3]. Already in the 1960s clinical and epidemiological data of a possible association between obesity and major depression were reported [4].

The incidence of depression in obese individuals is close to 30% [5,6], a rate that is significantly higher than that measured in the general population. Several studies have reported that being obese was associated with a risk of developing depression between 1.18 and 5.25, depending on the studies and evaluation methods [7,8,9]. A recent study found that obese individuals are 55% more likely to develop lifetime depression, while depressed individuals are 58% more likely to become obese than the general population [2].

Various studies have provided evidence of a two-way link between depression and obesity and the presence of one increases the risk of developing the other. Furthermore, there are strong reasons to believe that these conditions are interconnected through a vicious circle which reinforces each other through adverse physiological adaptations [7,10,11]. Overall, the size of the effect is higher if we consider high obesity (class III: BMI ≥ 40) compared to the application of the BMI cutoff of 30 that defines obesity and although positive they are less strong and not always significant for overweight (BMI 25–30). Sex has been shown to moderate the depression–obesity association since it was stronger in women than in men. Finally, several meta-analyses show that the depression–obesity association extends to bipolar depression, already exists in childhood and adolescence and is consistent in western and non-western countries [12,13,14,15,16]. Another study revealed that the onset of depression in early adolescence was associated with an elevated risk of late-onset obesity and obesity, particularly in late adolescence, was associated with an increased chance of developing depression in adulthood [17]. A recent study found a *U*-shaped relationship between body weight and depression, with a higher prevalence of depression in underweight and obese subjects compared to normal weight controls [18].

Obesity and depressive disorder are closely related, but the relationship is probably multifactorial and complex, focused not only on psychological and behavioral aspects, but also on shared biologic mechanisms that can explain the depression–obesity association at different levels, from genetics to mechanisms peripheral endocrines, from immuno-inflammatory and metabolic ones, to the involvement of the intestinal microbiota [10].

## 2. Psychological and Behavioral Aspects

From a cognitive and psychosocial point of view, obesity can strongly influence the self-image with a self-devaluation perception, social withdrawal, exclusion and social stigma, in particular when obesity is uncommon in their social networks, promoting the onset of a depressive symptomatology, particularly in a context of high social expectations and beauty standards. Conversely, depressive symptoms can help promote overweight and obesity due to an unhealthy lifestyle, such as excessive sedentary lifestyle, excessive alcohol consumption and unhealthy eating habits [19]. Psychological factors also play a major role in maintaining this link. For example, emotional eating (the tendency to eat in response to negative emotions) has been associated with both depression and obesity [20].

In addition, food preferences can change during times of stress or depression. A study conducted among British university students showed a positive correlation between depressive symptoms and consumption of highly palatable and high-calorie food, mainly high in carbohydrates, including sweets, biscuits, snacks and fast food [21]. In fact, high carbohydrate meals can temporarily improve mood, particularly because the consumption of highly palatable foods activates the brain opioid system, producing hedonic responses and stimulates the serotonergic system since the consumption of carbohydrates can increase the production of serotonin in the brain due to the increased availability of tryptophan [22,23].

In addition, mood disorders are often associated with changes in the sleep cycle that can induce changes in neuroendocrine functions with increased cortisol, dysfunction of glucose metabolism, increased levels of ghrelin (orexigen hormone), decreased levels of leptin (anorexigenic hormone) and, consequently, an increase in appetite [24].

## 3. Immuno-Inflammatory Aspects

Immunological and inflammatory aspects are also involved in the interactions between depression and excess weight. During a normal immune response, in fact, inflammation and adaptive behavioral changes occur only for a limited time. However, if these responses become exaggerated or prolonged over time, they can compromise different regions of the brain involved in the regulation of complex behaviors, not only in the control of food intake and the perception of satiety, thus promoting an energy imbalance, but may also have implications on cognition and on mood and strongly contribute to inactivity, weight gain and profound metabolic alterations [25,26,27].

Therefore, during an infection, the transient activation of brain cytokines coordinates a large number of behavioral changes (including weakness, listlessness, malaise, anorexia, fatigue and transient alterations of cognition and mood). These conditions are functional for recovery from infection, and usually resolve within days once the pathogens have been cleared and the innate immune system is no longer activated. However, the inability to effectively regulate systemic immune activation and/or cerebral microglial activation results in significant and prolonged hyperactivation of peripheral and cerebral cytokines. Such hyperactivation of the immune response in turn could culminate in medical conditions that adversely affect clinical outcomes, including neuropsychiatric symptoms, particularly when they ultimately affect key areas of the brain, such as the hippocampus, cortex or amygdala [27,28].

The inflammatory mediators network is represented by a remarkable array of molecules, the most important of which are proinflammatory cytokines (for example the interleukins IL-1β, IL-6) and the Tumor Necrosis Factor (TNF-α) produced within the innate immune cells in response to the immunological stimulus. Other anti-inflammatory cytokines oppose this response by attenuating the production of proinflammatory cytokines or by antagonizing their action at their receptor level (e.g., IL-1RA) In turn, the action of peripheral proinflammatory cytokines, such as on hepatocytes, leads to the synthesis of other acute phase proteins, such as the C-reactive protein (CPR), responsible for the systemic inflammatory response [29]. High systemic levels of these molecules, in the absence of infection or tissue damage, are considered abnormal and tend to support the onset, for example, cardiovascular diseases, diabetes, metabolic changes, neuropsychiatric diseases and ultimately mortality [30,31].

Both depression and obesity are disorders associated with the dysregulation of the stress system. The mechanisms linking these two conditions have been studied extensively indicating the involvement of alteration of the hypothalamic-pituitary-adrenal (HPA) axis, inflammation, oxidative stress and endocrine dysfunction [32]. Several scientific evidences suggest that there is a chronic inflammatory state in depression. The results that support this notion include evidence of an increase in proinflammatory cytokines in the depressive phase and, instead, of their reduction following the resolution of depressive symptoms due to the inhibitory effect on the production of cytokines by antidepressant treatments. In treatment-resistant depression there is an upregulation of proinflammatory cytokines [33,34].

Even in obesity, various evidences from the literature report that low-grade chronic inflammation is associated with it. This is characterized by an increase in circulating proinflammatory cytokines and accumulation of immune cells in different tissues and organs, including the central nervous system (CNS). During chronic stress there is evidence that these cytokines promote depression-like behavior by disrupting the synthesis of neurotransmitters and signal transduction. Animal models of obesity and depression revealed a two-way relationship whereby high-fat nutrition and chronic stress synergize and aggravate metabolic dysregulation and behavioral abnormalities. Although not yet exhaustive, several evidences from the literature suggest that inflammation in the central and peripheral nervous system can link obesity to major depressive disorder [35].

Obesity must be defined as more than just a metabolic disorder; there is in fact an evident state of alteration of systemic inflammation, so much so that today chronic low-grade inflammation is considered a hallmark of obesity. Because of their ability to act remotely on different organs, the inflammatory component can participate in the etiology of many of the metabolic complications associated with obesity [36]. Chronic obesity is often associated with hypertension, coronary heart disease, dyslipidemia, hyperleptinemia, reduced glucose tolerance linked to hyperinsulinemia and insulin resistance [28] and a greater susceptibility to immune-mediated diseases [37], and infections [38].

White adipose tissue (WAT), especially that located in the abdominal area, is a true endocrine organ that produces hormones, such as leptin and proinflammatory cytokines and therefore contributes significantly to the immuno-metabolic responses involved in genesis and maintenance of pathogenic conditions, such as obesity and depression [39]. In fact, due to the nutritional overload, the white adipose tissue becomes hypertrophic, with hyperplastic adipocytes which, due to this increase in volume, activate gene loci which induce the production of proinflammatory cytokines and favor the recall of macrophages by establishing an inflammation called the feed-forward process [29,40,41]. Substantially the inflammatory markers in obesity are more related to measures of abdominal adiposity, such as waist circumference and waist-to-hip ratio, rather than to the general measure of body mass index (BMI), which represents an approximation of fat total body and does not distinguish between high muscle mass and fat mass [26]. There is evidence that the association between depression and obesity is stronger for abdominal obesity. Abdominal obesity, characterized by the accumulation of visceral fat, is more strongly linked to metabolic and inflammatory dysregulations.

An interesting meta-analysis [42] confirmed that the association between abdominal obesity and depression was stronger than that between general obesity and depression. In contrast, weight loss, induced by low calorie diets or bariatric surgery, significantly reduces peripheral inflammation in obese individuals [43,44]. This peripheral immune activation, through both humoral and neural pathways, especially vagal ones, can induce an inflammatory brain state, which is associated not only with metabolic dysregulations, but also with emotional and behavioral alterations [45]. In obese individuals, high values of a wide range of proinflammatory cytokines can be found in circulation, including the Monocyte Chemoattractant Protein-1 (MCP1/CCL2), various interleukins such as IL-1β, IL-5, IL-6, IL-8, IL-12, IL-18, Interferon gamma (IFNγ), TNF-α and C-reactive protein (CRP) and many of these cytokines are implicated in metabolic inflammation and subsequent metabolic dysfunction [46].

An interesting meta-analysis [42] confirmed that the association between abdominal obesity and depression was stronger than that between general obesity and depression. In contrast, weight loss, induced by low calorie diets or bariatric surgery, significantly reduces peripheral inflammation in obese individuals [43,44].

For example, IL-1β, IL-6 and TNF-α have been shown to contribute directly to insulin resistance by activating stress kinases, such as IκB kinase (IKK), c-Jun N-terminal kinases (JNKs) and the P38 mitogen-activated protein kinases (MAP38) in muscle and fat cells, which inhibit the function of the insulin receptor (IRS1), thus blocking signal transduction [47,48]. IL-12 and IFNγ cytokines play key roles in the activation of the immune system; IL-12 promotes the cytotoxic differentiation Th1 of CD4 + T cells and the IFNγ promotes the activation of class M1 proinflammatory macrophages [49]. Furthermore, chemokines such as CCL2 and CXCL1 are able to induce chemotaxis, causing immune cells to escape from the bone marrow and subsequently migrate to the tissues [46,50].

In metabolic syndrome or obesity, the total monocyte, neutrophil and lymphocyte count increases and correlates positively with body mass index (BMI), body fat percentage and insulin resistance [51].

Another distinctive feature of metabolic inflammation linked to obesity, is the infiltration of immune cells into the tissues that regulate glucose metabolism throughout the body. This phenomenon was initially observed as an increase in total macrophages in the adipose tissue of obese patients [52], but subsequent studies have found that T cells, B cells, eosinophils, mast cells, Natural Killer cells and neutrophils can infiltrate adipose tissue and contribute to the regulation of insulin sensitivity [46]. Macrophages that accumulate in tissues during obesity are largely derived from monocytes in the tissues dependent on the chemokines CCL2 and CCR2 [53]. Furthermore, saturated fatty acids, a substantial component of obesogenic and atherogenic diets, but not unsaturated fatty acids, are able to directly activate the proinflammatory pathways in macrophages. Exposure of adipocytes, hepatocytes and myocytes to excess saturated fatty acids or inflammatory stimuli can directly induce insulin resistance [46]. Moreover, obesity is associated with various metabolic disorders that can lead to an increase in cortisol, leptin and insulin levels, with consequent dysregulation of the HPA axis and insulin resistance which can further induce inflammation and worsen depression [32].

The link between depression and inflammation was initially suggested by clinical results showing that depression is accompanied by an upregulated inflammatory response, such as increased production of proinflammatory cytokines and acute CRP. The link between inflammation and depression is probably two-way [54]. Interferon (IFN-α) immunotherapy has been shown to precipitate depression, even in people without any psychiatric history. Up to 45% of patients receiving IFN-α develop depressive symptoms unless they receive prophylactic antidepressant treatment [3,55]. Similar to what is observed in patients with obesity, individuals with depression show low-grade chronic inflammation, which can be characterized through the profiles of circulating cytokines. Numerous studies conducted on patients meeting the criteria of the Diagnostic and Statistical Manual of Mental Disorders (DSM) [56] for major depression have found significant increases in plasma or serum levels of CCL2, IFNγ, IL-1α, IL-1β, IL-2, IL-6, IL-8, IL-12 and TNF-α, together with CRP [46,57]. The “cytokine hypothesis of depression” postulates that these cytokines play a causal role in the progression of depression [58].

A study found that, in treatment-resistant patients with major depression, administration of the TNF-α antagonist, infliximab, alleviated depressive symptoms in subjects with elevated basal inflammatory markers [59]. Several large meta-analyses report higher levels of inflammatory markers in depressed people than controls [60,61,62,63]. Recently, genome-wide association studies (GWAS) have identified significant associations between groups of genes involved in regulating cytokine synthesis and in the immune response with depression [10,64].

Different mechanisms have been proposed by which, within the brain, the various cytokines and immune cells could influence behavior and mood. The levels of cytokines such as IL-1β, IL-6 and TNF-α rise in the brain during conditions of stress or chronic depression and derive either from local production, in the central nervous system or from translocation through the blood–brain barrier (BBB) from the periphery [65]. They can directly access the brain by crossing the blood–brain barrier through a saturable active transport system or through indirect pathways including the activation of glial cells, in particular of microglia, within the central nervous system (CNS), through leukocytes [66]. While the activation of microglia normally exerts a protective action on the central nervous system, its unregulated and chronic activation can, on the contrary, become harmful. Within the brain, proinflammatory cytokines activate the neuroendocrine system, impair the metabolism and function of neurotransmitters and alter neural plasticity and brain circuits [58,67].

Activated microglia synthesizes IL-6 and TNF-α, as antineurogenic signals, which can interact directly with neural progenitor cells and determine a decrease in neurogenesis also on the brain structures that regulate emotions in depression [29]. The pathophysiology of depression is characterized by the alteration of the neurotransmitter modulation of some monoamines, such as serotonin (5-HT), dopamine (DA) and norepinephrine (NE). It has been shown that inflammatory cytokines, including IL-6 and TNF-α, together with other inflammatory factors, are able to induce the synthesis of the enzymes indoleamine 2,3-dioxygenase (IDO) and GTP-cyclohydrolase 1 (GTP-CH1) in monocytes/macrophages and dendritic cells, with consequent significant alterations in the biosynthesis of key monoamines (e.g., serotonin and dopamine) which play an important role in mood regulation and cognitive function. In addition, IDO is the first enzyme that limits the rate of catabolization of tryptophan along the quinurenin pathway, a path that ultimately leads to the production of neuroactive metabolites, 3-hydroxyquinurenine and quinolinic acid, which are associated with anxious symptoms and depressive [68].

Furthermore, the activation of the IDO also leads to an increase in the production of glutamatergic metabolites, which are known to induce neuronal death [26]. Interestingly, the hippocampus plays an important role in these phenomena; dysregulated activity of the hippocampal microglia has been associated with sustained IDO activity and therefore with protracted depressive behavior [26]. Furthermore, in mouse models, the emotional alterations related to the activation of the hippocampal IDO, induced by inflammation, have been associated with a reduced hippocampal expression of the brain-derived neurotrophic factor (BDNF). BDNF plays an important role in synaptic plasticity and neuronal survival in the hippocampus and other brain regions implicated in mood regulation and learning [68]. Furthermore, decreases in expression and mutations in the BDNF coding gene have been associated with obesity in human and animal models [69].

Overall, these results indicate a fundamental role in the activation of the IDO, in particular in the hippocampus, in mediating the mood and cognitive alterations induced by the various cytokines. There is also a growing literature suggesting that cytokines are able to act directly on neurons through cytokine receptors within the plasma membrane to modify excitability, connections and synaptic remodeling [70,71]. Furthermore, cytokines such as IL-1β may contribute to a greater activation of the HPA axis, thus aggravating the inflammatory response to stress [46]. Recent work has suggested that microglia also exhibit increased phagocytic activity during chronic stress, which may be involved in synaptic remodeling [45]. Taken together, these studies show the different multiple pathways by which chronic stress, through the activation of the immune system, can promote depressive behavior.

In addition, it should be added that in the regulation of inflammatory activation, the inflammasome has gained increasing interest in recent years. The inflammasome acts as a molecular platform in which a group of enzymatic protein complexes, induced by stress, breaks up the inactive forms of the proinflammatory cytokines IL-1β and IL-18, into biologically active forms, through the activation of a protease with cysteine in the active site, called caspase-1 [72]. Activation of the inflammasome is a crucial point for the defense of the organism from pathogens. Caspase-1 expression in inflammasome was shown to be upregulated in the adipocytes of obese patients [73] and that, instead, inhibition of caspase-1 can reduce weight gain in animal models, such as in mice with induced obesity [74]. Caspase-1 expression was increased in peripheral blood mononuclear cells from depressed patients [10,75]. In addition, the over regulation of caspases in the inflammasome can determine a protein cleavage in the glucocorticoid receptor (GR), compromising its reactivity and therefore contributing to the chronic activation of the HPA axis [10].

Given the bidirectional link reported between obesity and depressive symptoms, it is highly probable that the depressive symptoms that occur in the context of inflammation linked to obesity may in turn contribute to the maintenance of obesity, thus promoting the establishment of a circle vicious.

## 4. Alterations of Neuroendocrine Function

Alterations in neuroendocrine function represent another common feature in inflammatory conditions, including obesity and depression. The immune system and the neuroendocrine system are in constant communication and immune alterations are known to cause significant changes in neuroendocrine activity and vice versa it is highly possible that obesity-related neuroendocrine dysfunction contributes to neuropsychiatric comorbidity in obese individuals [26]. In general, it is known that depression is associated with impaired function of both the hypothalamic-pituitary-adrenal (HPA) axis, the immune system (inflammation) and the metabolic pathways. Furthermore, several studies have suggested that the neuroendocrine signaling processes that regulate both mood and energy metabolism are strongly interconnected [76]. In particular, obese subjects have been shown to have a reduced feedback response to cortisol, similar to that observed in depression [76].

## 5. Hypothalamic-Pituitary-Adrenal Axis (HPA)

The cerebral effects of cytokines on mood regulation and cognitive function are probably modulated by the close interactions existing between the inflammatory and neuroendocrine system, in particular with the HPA axis, which is significantly activated in obesity. Immune alterations are in fact known to cause significant changes in the activity of the HPA axis and vice versa [28,45]. As a rule, during stressful events, activation of the hypothalamic-pituitary-adrenal axis causes the release of glucocorticoids, such as cortisol from the adrenal glands, into the bloodstream [77]. Although glucocorticoids are normally immunosuppressive, chronic stress is hypothesized to stimulate hyperactivity of HPA, inducing resistance to glucocorticoids, which in turn causes proinflammatory activation of immune cells [78]. Long-term exposure to cortisol leads to neuronal damage and loss in limbic regions vulnerable to stress and associated with depression, such as the hippocampus and the amygdala [10,79]. Interestingly, glucocorticoids have recently been shown to sensitize microglia in an animal model of obesity. Indeed, mice with diet-induced obesity (DIO) show an exacerbation of HPA axis activation in response to an immune threat, along with an increase in neuroinflammation and depressive-like behavior [28,80].

A natural pattern of prolonged mood exposure to cortisol is Cushing’s syndrome (CS), characterized by endogenous hypercortisolism caused by pituitary or adrenal adenoma or bilateral hyperplasia of the adrenal cortex, which reverses after surgical removal or other targeted treatments hypercortisolism. Major depression occurs in 50–80% of patients with CS with active disease [81]. Importantly, the onset of depressive symptoms in Cushing’s syndrome and their improvement after treatment for hypercortisolism demonstrate a causal role of cortisol in depression. It should also be noted that exposure to high levels of cortisol can also induce obesity through various mechanisms: (a) increased appetite with a preference for energy-dense food; (b) promotion of adipogenesis and adipocyte hypertrophy, especially in visceral fat; (c) suppression of thermogenesis in brown adipose tissue (BAT) with relative reduction of energy expenditure [82]. It is conceivable that these obese patients with hypercortisolemia may be more prone to metabolic sequelae of obesity and depression. Chronic inflammation typical of obesity can severely limit the functioning of the glucocorticoid receptor (GR) and this reduced binding activity with circulating cortisol reduces the triggering of negative feedback and therefore does not sufficiently suppress HPA activity. In fact, proinflammatory cytokines activate the elements of the cell transduction cascade which prevent the nuclear translocation of GR or interfere in the interaction of GR with the elements of response to gene promoters [10]. Dysregulation of isoenzymes 1 and 2 of 11-β-hydroxysteroid dehydrogenase (11-βHSD), which catalyze the conversion of inert 11 keto-products (cortisone) to active cortisol and vice versa, thus regulating the access of glucocorticoids to steroid receptors causes alteration of cortisol metabolism which is often present in both obesity and depression [83]. Furthermore, a reduced activity of 5α-reductase, which induces a reduction in the clearance of glucocorticoids, can enhance the accumulation of visceral adipose tissue and influence the development of a depressive symptomatology [84]. Finally, the activity of liver enzymes responsible for cortisol clearance and regeneration has been shown to be altered in patients with nonalcoholic fatty liver disease (NAFLD), which is one of the typical metabolic sequelae of abdominal obesity [10].

## 6. Leptin

This adipokine has been extensively studied in recent decades for its key role in controlling energy homeostasis and eating behavior. The leptin-melanocortin pathway is a key neuroendocrine regulator of energy homeostasis. Leptin is produced from white adipose tissue in proportion to body fat and acts as a signal of abundance of adiposity at the central level as it binds to receptors expressed on neurons of the hypothalamic nuclei that promote the release of proopiomelanocortin (POMC). POMC is a prohormone, which with suitable cuts (carried out via proconvertase), originates various peptides from melanocortin (i.e., α, β and γ MSH) [76] which interact with other hypothalamic nuclei of the feeding area to integrate physiological processes and behavioral patterns that suppress food intake and promote energy expenditure [85]. In addition to controlling food intake, leptin modulates sexual maturation, reproductive functions, immune functions and the HPA axis through negative feedback on the hypothalamus. Leptin is secreted in a pulsating manner and its secretion is inversely related to that of ACTH and cortisol. Furthermore, due to the wide cerebral distribution of leptin receptors, which are also found throughout the cortex and hippocampus, leptin has also been shown to modulate memory processes and mood disorders [86,87]. The most common forms of obesity are associated with leptin resistance (a process similar to insulin resistance in type 2 diabetes), which mitigates its anorexigenic effect and consequently inhibits nutrition, despite the high circulating leptin. Central leptin resistance is due to impaired transport of leptin through the blood–brain barrier, reduced function of leptin receptors and defects in the transduction of its signal [88].

Inflammation linked to obesity plays an important role in altering, centrally, the action of leptin. For example, CRP has been shown to directly inhibit the binding of leptin to its receptors. Furthermore, central inflammation can compromise the activity of the hypothalamic leptin receptor level by activating inhibitory signals from multiple negative feedback circuits [88].

The effects of leptin on mood can be exercised through various mechanisms: direct action on the receptors of neurons present in the hippocampus and amygdala, enhancement of neurogenesis and neuroplasticity in the hippocampus and in the cortex and modulation of the HPA axis and the system immune [10] by regulating the activation of peripheral immune cells and brain microglia [28,36]. These results suggest that both leptin and cytokines may contribute together to the development of behavioral changes associated with obesity.

## 7. Insulin

Insulin, whose circulating levels and signaling pathways are often altered in obesity, is also able to interact with inflammation processes and act not only on peripheral tissues, but also on insulin receptors present in brain, in particular, in the hypothalamus, deputies for energy control, glucose homeostasis and eating behavior [3,89]. Furthermore, at the molecular level, the presence of inflammatory cytokines (in particular IL-1β and TNF-α) have shown that they can compromise the effectiveness of the insulin receptor in signal transduction, not only at the peripheral level, but also at the inside the brain [90,91]. The impaired insulin signaling pathway may, therefore, as with leptin, contribute to the development of neuropsychiatric symptoms in the context of obesity. In addition, the inflammation condition present in obese patients, with increased concentrations of proinflammatory cytokines and high presence of macrophages in adipose tissue can have a significant impact on the functioning capacity of insulin, not only by reducing the secretory function of the β cells pancreatic up to apoptosis, but also by attenuating the insulin receptor’s ability to propagate downstream transduction [10,92] and promote the condition of insulin resistance [93]. The alteration of brain metabolism, due to insulin resistance, has been associated with impaired memory and executive functions and neuronal damage, both in the hippocampus and in the medial prefrontal cortex [94]. Therefore, insulin dysregulation has been hypothesized to play a role in neuropsychiatric conditions such as dementia and depression [94]. A small, but significant cross-association between depression and insulin resistance was found in a large meta-analysis involving 21 studies [95]. Furthermore, some meta-analyses underline the frequent association between depression and type 2 diabetes mellitus (T2D) in a substantially bidirectional way [96,97]. Insulin dysregulation can probably represent a mediating mechanism in the obesity–depression relationship, strongly influenced by environmental factors [10].

## 8. Microbiota

By microbiota we mean the set of symbiotic microorganisms that coexist with the human organism without damaging it while the term microbiome refers to the genetic heritage of the microbiota. In humans there are several million different species of microorganisms, the most numerous of which are bacteria, but also, albeit to a lesser extent, fungi and viruses [98]. Among the bacteria the majority is anaerobic, more or less narrow or optional (many survive in the absence of oxygen and some tolerate its presence). The total number of genes in the microbiota is estimated to be one hundred times the number of genes in the human genome. A large part of human genes is acquired by the microbiome present in the body [99].

It is now believed that the microbiota should be considered as a real organ, an immune-metabolic organ that performs functions that we would otherwise not be able to perform. These functions include the ability to assimilate otherwise indigestible components of our diet, disrupting substances that our body is unable to dismantle, such as cartilages and vegetable polysaccharides or to synthesize essential substances, such as vitamin K, which plays a role essential in blood clotting. Furthermore, it has been observed that the microorganisms that colonize the gastrointestinal tract are active protagonists of intense interactions between the gastrointestinal tract and the neuro-immuno-endocrine system [100,101].

It is therefore necessary to consider the substantial importance of the coexistence in our organism of the microbiota and evaluate the plastic dynamism of the microbiome, capable of transforming its gene expression in relation to environmental factors such as the type of diet and impact on the state of health. It has been observed that individuals from the same family have a similar core of classes and species of intestinal bacteria that can change based on interactions with the host and the environment. Three main bacterial phyla are recognized in the normal weight individual: Firmicutes, Actinobacteria and Bacteroidetes [102,103]. Certain factors such as diet, drugs, the presence of comorbid diseases have been implicated in changes in the composition of the intestinal microbiota and, consequently, in the possible development of metabolic and neuropsychiatric disorders [104].

A relationship between intestinal microbiota and obesity was also showed. Some studies, carried out both in mice and humans, have shown a change in the composition of the intestinal microbiota in obese subjects with an increase in Firmicutes and a reduction in Bacteroidetes. In essence, the differences in the extraction of calories from substances ingested with food can be largely dependent on the composition of the intestinal microbiota and, at the same time, weight loss is able to restore the normal intestinal microbial composition, confirming the link between the microbiota and obesity [105,106,107].

Several studies show an impact of the intestinal microbiota on weight gain, comparing axenic mice (without microbiota) to conventional mice: the latter are capable of better digesting dietary fiber and extracting more energy than axenic mice. Microbiota are therefore an important “organ” that allows to guarantee the normal functions of the intestine. Fecal microbiota transplantation (FMT) from conventional mice to axenic mice led to an increase in their fat mass, which proves the impact of the microbiota on weight gain. It has also been shown that the FMT of obese and lean mice to receiving axenic mice caused greater weight gain in mice that had received the obese mice microbiota. Another similar experience, but this time with human microbiota, showed that mice that received the microbiota through FMT in obese patients gained more weight and increased their fat mass more than the mice that received the microbiota from lean patients instead. The composition of the microbiota is therefore important, but also its diversity. Having a “poor” bacterial intestinal flora may be associated with obesity. The weak quantity of some bacteria or the absence of these could also be a risk factor that affects obesity [108,109,110].

Therefore, the microbiota can affect the body’s nutritional and metabolic balance by modulating the ability to extract energy from diet foods and interacting with the glyco-lipid metabolism [102]. The metabolites released by the fermentation of complex diet polysaccharides can increase the absorption of glucose, stimulate lipogenesis, modify the fatty acid composition of the fatty tissue and the liver, alter the permeability of the intestinal mucous barrier, alter the immune response, contribute to a state of chronic systemic inflammation (metabolic endotoxemia) and a state of insulin resistance related to obesity [104,105,106,107].

Interestingly, gut microbiota transplantation from mice with obesity induced to lean mice has recently been reported to be sufficient to induce both microglial activation in the brain and neurobehavioral changes in the absence of obesity [111]. This elegant study supports the idea that intestinal microbiota alterations related to obesity can modulate the gut–brain communication pathways, leading to the development of neuropsychiatric comorbidities associated with neuroinflammation. Similar to this hypothesis, the use of compounds that improve the microbiota (for example prebiotics or probiotics) appears to be a promising way to improve neuropsychiatric comorbidities in obese patients [112,113]. More generally, nutritional interventions based on factors with immunomodulating properties, in particular omega-3 polyunsaturated fatty acids and antioxidants, have proved to be possible good strategies for the development of new therapies for neuropsychiatric disorders related to obesity [114].

Numerous data have shown that depression is associated with an altered composition of the gut microbiota, generally in the form of reduced wealth and diversity [115,116,117,118,119]. As in obese patients, also in depressed patients, an increase in the translocation of intestinal bacteria is detected which, by passing the intestinal mucosa, favors the activation of immune responses [120]. The excessive presence of lipopolysaccharide (LPS), a substance present in the outer membrane of bacteria, can cause metabolic endotoxemia that activates systemic macrophages through the binding of LPS to its specific receptor which triggers the immune system by inducing an inflammatory response [23,28]. On the contrary, after weight loss in obese individuals, reduced serum levels of the LPS-binding protein, a marker of endotoxemia, were found [28,121]. As already pointed out, changes in the intestinal microbiota in general are relevant to the mood because the microbiota interacts with the brain through neuro-immune, neuroendocrine and neural pathways [101]. Communication from the intestine to the hypothalamus is also mediated through the HPA axis and it is assumed that communication, through this axis, is bidirectional, with the intestine being able to send return signals to the brain [115,122]. To date, however, the greatest available evidence shows that the main route for this signaling is through the nervous system, in particular through the vagus nerve. The vagus nerve is an important signal transducer from the brain to the viscera, however, approximately 80% of the vagus nerve fibers are afferent, transmitting sensory information from the viscera, including the digestive tract, to the brain for integration and appropriate responses. to maintain homeostasis [115,123]. Intestine–brain communication can also be indirect or mediated by different metabolites. For example, the intestinal microbiota can influence brain states by modulating the production of neuroactive substances such as serotonin, norepinephrine, dopamine, glutamate and gamma-aminobutyric acid (GABA) [124]. Obviously, the intestinal microbiota can alter the functioning of the brain indirectly through the change of the inflammatory and immune states.

The growing interest in this area of research will undoubtedly lead to greater insights into the mechanisms underlying microbiome–gut–brain communication and will provide a new understanding of the potential of microbe-based therapeutic strategies that can help treat mood disorders [125]. The microbiome topic is so relevant that in the USA the National Health Service (NHS) is carrying out the Human Microbiome Project, a project with a total budget of 115 million dollars, which aims to identify and characterize microorganisms and their relationship with the state of health and disease of man [106] 

## 9. Genetic Aspects

Various authors, in very recent articles, have highlighted close links also of genetic susceptibility between depression and obesity [126,127]. In a 2019 study, researchers examined genes associated with obesity, but not associated with metabolic diseases, and found that, in obese and depressed patients, several genes were as frequent as genes that determine both obesity and disease metabolic, such as diabetes. They therefore highlighted that weight gain, even when not associated with other diseases, is associated with an increased risk of developing depression. Research suggests that people with genetic variants linked to a high body mass index (BMI) would be more likely to suffer from depressive syndromes related to psychological factors. According to the authors, genetic variants linked to a high body mass index (BMI) can lead to depression, with more evident effects on women than on men. By focusing on the 73 genetic variants linked to a high body mass index and a higher risk of metabolic diseases, and taking into account factors such as age and gender, the researchers found that for each 4.7 point increase in BMI, the probability of being depressed increased by 18% overall and 23% among women. Overall, the team of researchers found that participants with a higher body mass index were more likely to be depressed. The results remained the same even in the additional tests that excluded people with a family history of depression, even when the analysis was repeated using the data of another large international project called Psychiatric Genomics Consortium and therefore it is suggested that the psychological component it is as strong as the physiological one, if the latter is present. Although the study has some limitations (it mainly concerns white people of European origin and some data are reported by the patients themselves), it concludes that of course, many other factors can cause depression, but weight loss could still be useful for improving mental health in some individuals, and staying thinner in general can help reduce the chances of developing depression [128].

## 10. Conclusions

Depression and obesity are currently important public health concerns due to their growing prevalence worldwide, their important impact on health and morbidity and the massive social and economic cost. Many clinical evidences point out an intricate and complex relationship that leads to the conclusion that depression and obesity can interact with each other in a bidirectional longitudinal association. Furthermore, at the clinical level, the simultaneous presence of depression and obesity determines a significant aggravation of the conditions in the individual patient and has important clinical implications as this comorbidity can represent a serious obstacle in the treatment of each condition taken separately. Indeed, in depressed patients, biologic dysregulations related to obesity are often associated with a longer course, a worse prognosis [10] and a reduced response to standard antidepressant treatments [129]. Similarly, the presence of depression in obese patients can significantly reduce adherence to treatments for obesity and its complications through less adherence to pharmacological and lifestyle prescriptions [130].

The results of several studies presented in this review support the hypothesis that the condition of inflammation is the main and crucial mediator of the relationship between adiposity and depression and which moreover involves other systems, such as the immune, the neuroendocrine one, in particular the HPA axis, the gut microbiota and key areas of the brain, including the hypothalamus, hippocampus and basal ganglia. The consequent alterations in the metabolism and function of the monoamines, the altered activity of the neurotransmitters together with the occurrence of neurotoxic effects can favor the reduction of neurogenesis and neuronal death which represent the main pathogenetic pathways of neuropsychiatric morbidity in obese individuals [26].

Central and systemic inflammation, therefore, assumes the role of link between the psychological and biologic determinants that interact between obesity and mood. In addition, alterations in the intestine–brain axis represent a mechanism of neuropsychiatric comorbidity that can be induced by inflammation and that can be relevant for both the condition of obesity and that of mood disorders. As previously mentioned, obesity is associated with alterations of the intestinal microbiota in the form of changes in its population, increased permeability of the intestinal mucosa and activation of inflammatory processes. There is a rich and complex communication network between the intestine and the brain that involves both the endocrine, immune and neural pathways, and there is now multiple evidence that compromise or dysregulation of the intestine–brain axis affects mood and on cognitive function. These data suggest that alterations of the intestinal microbiota found in obesity may modulate the gut–brain communication pathways, thus leading to the development of neuropsychiatric comorbidity [28]. In fact, the gastrointestinal tract is the largest immune-endocrine organ in mammals, secreting dozens of different signaling molecules, including peptides. Peptides released by specialized cells in the gut participate in gut–brain communication. There is a significant anatomic and functional overlap of the peptides released in the intestine and brain, suggesting that these peptides exert common downstream effects on the neural systems involved in mental health. Intestinal peptide concentrations are not only modulated by the enteric signals of the microbiota, but also vary according to the composition of the population of the intestinal microbiota. Intestinal peptides in the systemic circulation can bind to receptors present on immune cells and on the terminals of the vagus nerve, thus allowing indirect intestine–brain communication [115,131]. Clarifying these mechanisms that link metabolic alterations, depression and inflammation could generate potential new therapeutic targets or specific strategies to combat both obesity and depression (Figure 1) [35].

The associations between obesity, inflammation and depression seem robust, however, there is some evidence to suggest that this may be particularly true of “atypical” depression, which is a fairly common disorder. According to DSM 5, it shares several characteristics with major depression, but it differs in some more evident specific symptoms such as biologic and vegetative ones. The patient mostly complains of physical pain, tiredness and physical weakness. These people also have other atypical symptoms: the “leaden paralysis” or a feeling of heaviness and tiredness, like feeling the arms and legs of lead, with intense tiredness that concentrates in the extremities, up to frequently experiencing pain and evident difficulties mobility; hypersomnia or hyperphagia or sleeping and eating excessively, respectively. Women with atypical–but not melancholic–depression were more likely to have a higher fat mass than controls [132] and in the elderly with depression, those with atypical forms had the greatest metabolic dysregulation [133]. More recently, the PsyColaus study has also provided evidence for a longitudinal link between atypical depression and overweight/obesity and metabolic syndrome [134,135].The association of immune-metabolic dysregulations, including chronic low-grade inflammation with proinflammatory cytokines, oxidative stress, alterations of neuroendocrine regulators (leptin and insulin resistance for example) or biomolecular (dyslipidemia), related to metabolism energy and underlying genetic vulnerability, may be present not only in obesity, but also as common—or even specific—conditions in the atypical depressive subtype [10].

To conclude, it can be highlighted that an increased appetite, reduced mobility and hypersomnia in the context of a depressive episode can denote a greater association between depression and metabolic alterations and the presence of markers of inflammation. Appetite regulation could be an important factor in the beginning of an immune-metabolic form of depression. Among other things, many of the comorbidities associated with obesity and depression such as metabolic syndrome, cardiovascular diseases, diabetes and some tumors share a background of greater inflammatory activation. Individual variability can be related to psychosocial variables that can amplify genetically determined biologic vulnerability [136]. It is also important to recognize that various biologic mechanisms examined can also be influenced by various individual behaviors, including smoking, alcohol consumption, sedentary lifestyle, poor nutrition and socioeconomic status [137,138,139].

## Figures and Tables

**Figure 1 diseases-08-00023-f001:**
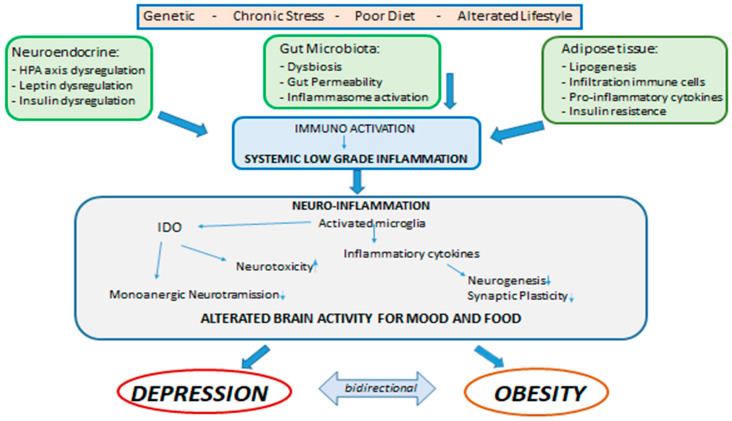
Depression and Obesity: interaction between genetic factors, chronic stress, unhealthy diet and lifestyles.

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
