# Peer review of "Depression and Obesity: Analysis of Common Biomarkers"

_diseases, 2020, doi:10.3390/diseases8020023_

Round 1

Reviewer 1 Report

This is an interesting paper that discusses the association between obesity and depression, presenting the probable mechanisms involved. I have some concerns as follows, though:

  1. Review articles would benefit greatly from figures that summarize the contents, particularly when mechanisms are discussed.
  2. Please go through the manuscript and correct many typos (e.g. hormone hormone in line 84; line 316, etc.).
  3. Emotional eating might be a better word than emotional food in line 72.
  4. Line 105: IL-1beta?
  5. Line 108: IL-10 is a regulatory cytokine, not anti-inflammatory.
  6. Lines 124-134 are not in English!!!
  7. Line 152: total body fat is correct.
  8. Lines 168-178 are repeated!
  9. Line 207: what do you mean by PCR?
  10. Line 313; what is mean by prolonged mood exposure to cortisol?
  11. It is really beneficial that you have mentioned the difference between melancholic and atypical depression in terms of obesity and inflammation. Maybe you should include this important point in the abstract as well.
  12. Antidepressants usually cause weight gain too. In those with melancholic depression, the medications improve appetite and can lead to obesity in long term. Please verify this and include it in your paper as appropriate.
  13. As for obesity, there is some evidence that fecal microbial transplantation (FMT) from depressed patients to animals, causes depression. This could be mentioned in section 8 to further strengthen your discussion.
  14. The conclusion section is more of a summary than a conclusion. Perhaps the current conclusion section could be split into a summary section and then a proper conclusion section that synthesizes rather than repeats.

Author Response

We are very grateful to reviewers for the useful suggestions

According to the suggestions of the two referees, the following changes were performed: all corrections are highlighted in green:
- The various suggested lexical, conceptual and translation corrections were revised as requested
- A full reference has been added to the possible strategies of fecal microbiomal transplantation (FMT) with related bibliographical references
- A chapter has been added on the genetic aspects between depression and obesity with related bibliographical references
- Figure 1 is present in the body the text.

Reviewer 2 Report

This is a good review on an interesting topic. However, the genetic aspect of depression and obesity is missing. Suggest the authors to look into that aspect. 

also, line 518 has FIG. 1 in it. but I cannot see any figure. 

All other parts looks fine. 

Author Response

(The authors gave the same response as above.)

Round 2

Reviewer 2 Report

I am satisfied with the revision